# Effect of Indoor Climbing on Occupational Self-Efficacy and Employability: Results of a 10-Month Randomised Controlled Study of Persons with Intellectual Disability

**DOI:** 10.3390/ijerph192013399

**Published:** 2022-10-17

**Authors:** Ruud Joseph Alida Vreuls, Jonas Mockenhaupt, Vera Tillmann, Volker Anneken

**Affiliations:** FIBS gGmbH, Research Institute for Inclusion by Physical Activity and Sport, Associated Institute with the German Sport University Cologne, Paul-R.-Kraemer-Allee 100, 50226 Frechen, Germany

**Keywords:** indoor climbing, intellectual disability, RCT, employability, occupational self-efficacy

## Abstract

(1) Background: Indoor climbing has different effects at various levels, including physical, psychological, and social ones. It is of high interest to assess whether social skills developed through climbing can be transferred to another environment, such as the working environment. This study investigates the effects of indoor climbing on employability and occupational self-efficacy of people with intellectual disability, who possess lower levels of social competences in general. (2) Methods: A randomised controlled study (RCT) experimental study design with three groups was formed—one intervention (IG) and two control groups (CGI&II). For 10 months, the IG went climbing (two times per week), whereas the first CG followed a sports programme and the second CG served as controls. (3) Results: IG participants showed significant improvement in mental and somatic health over time. Regarding occupational self-efficacy, females had a significantly lower mean. Nevertheless, only the IG’s female participants mean increased significantly over time. (4) Conclusions: Indoor climbing can be effective for improving occupational health and can be beneficial for specific groups; however, additional research is needed to further specify the influence of indoor climbing on a wider variety of aspects of the life of people with intellectual disability.

## 1. Introduction

Over recent years, climbing as a sport has gained much popularity. Due to this, climbing is one of the new sports that has been added to the latest summer Olympics Games [1]. There are several positive aspects about climbing, which can be seen as reasons why climbing has become very attractive as a sport over the last couple of years. First, it can be performed in different ways (e.g., the three Olympic subdisciplines of bouldering, speed climbing, and lead climbing) and can therefore fit individual interests. Furthermore, the risk of injury while performing sports climbing (basically lead climbing and top rope) is very low [2,3,4]. Last, the requirements necessary for sports climbing are quite uncomplicated, since only a small amount of equipment (e.g., shoes and a harness) is required. Thus, climbing is a sport that can easily be adapted to everyday life. Therefore, observing the impacts on work performance when climbing is integrated into the daily routine over a longer period is of great interest, in addition to the possible effects climbing can be expected to have in the long term on one’s own ability, such as employability and occupational self-efficacy.

Participation in working life is of particular importance in order to enable participation in general life. A key factor for this purpose is employability, which is defined as the individual’s potential to take up, maintain, and expand employment [5]. One of the critical factors for good employability is social stability, which, for example, could include a person’s network and their direct social environment. Sheltered workshops make an important contribution to social stability: people with disability develop a regulated daily structure and social contacts, which give them a sense of belonging, thus ensuring participation in working life [6]. According to the current figures of the German association of sheltered workshops [7], more than 312,000 people with disability are employed in sheltered workshops in Germany, of whom more than three quarters have an intellectual disability. The main interest of this association, also known as BAG WfbM, is to support occupational participation in daily life. Findings from the report of the German employment agency show that the employability rate for people with disability was 46.9% and for people without disability—75.2% [8] (p. 7). These findings regarding people with disability need to be analysed somewhat critically, since this particular group includes people with different kinds of disability. Additionally, the report showed that the employability of people with disability rose in terms of absolute numbers, although the overall percentage of employment for people with disability dropped [9] (p. 8). Furthermore, data from the participation report of the German government regarding current employment indicate that the unemployment rate is higher for people with disability (13.4%) than that of non-disabled people (8.2%) [7].

Several studies investigate the unemployment rate in the general labour market of people with disability in particular, which continually remains high over several years in comparison to the unemployment rate for people without disability [8,10,11]. For example, it is stated by the German employment agency [8] that the challenge of the labour market of people with disability in Germany is merely caused by the legal framework and demographic development rather than by the economy. Furthermore, Ref. [12] also state that the participation of people with disability in the general labour market is lacking, since the transition rate from the sheltered workshop is currently below 1%. Ref. [13] conducted a study about the absence of people with disability working in the hotel industry. Several reasons for this finding were stated, such as them *lacking the required capacities* and the *high costs of accommodating and training people with disability*. The study outcomes showed that these reasons turned out to be negative judgements towards people with disability, as Ref. [13] found that these judgements lacked an overall foundation in facts. These findings support a statement in [14] that people with disability are often categorised as a ‘problematic group’.

Contrary to the reasons mentioned before—which are stated from employers’ perspectives—there are also factors influencing employability, which are directly from the employee’s perspective. Ref. [15] listed factors that influence employability, such as: skills and capabilities, somatic and mental health, activity level for job search, and further education. Self-efficacy can be seen as a factor in influencing employability. Ref. [16] describes self-efficacy as the belief in one’s own abilities, specifically the ability to meet challenges and complete a task successfully and state that self-efficacy is essential in the working environment. The meaning of self-efficacy in the working environment can be further defined as occupational self-efficacy. The essence of occupational self-efficacy is dependent on the capability of one’ senses towards their overall work-related competence [17]. Furthermore, the construct of occupational self-efficacy correlates positively with both commitment and job satisfaction [17]. In addition, Ref. [18] carried out a meta-analysis regarding the influence of occupational self-efficacy and also showed that it influences employees’ performances.

Both [14,15] described one’s health status as one of the strongest factors in influencing employability. The impact of physical activity on health, therefore, is very comprehensive: it affects the cardiovascular system, metabolic processes, diabetes (in a positive way), and also mental health characteristics, for instance, subjective well-being and cognitive performance [19]. It is evident from the abovementioned information that sports can be very valuable to one’s health. However, only a few of them can be beneficial to both physical and mental health. Climbing can be categorised as one of these few sports. Hardly any studies can be found that focus on the psychological effects of climbing. Ongoing research is being conducted in Germany to evaluate the effects of bouldering therapy in combination with occupational therapy on depression. The authors of [20,21] evaluated the effect of climbing on acute emotion regulation in comparison to relaxation sessions with persons with major depressive disorders. In a non-randomised pilot study, 40 people were divided into a climbing group and a relaxation group. The results showed the positive significant effects of climbing on coping with emotions, negative affect, and depressiveness in contrast to the relaxation group. Furthermore, as stated in [22], the execution of movements in climbing are far more complicated in comparison to sports such as swimming, running, and cycling. This is mainly caused by cognitive demands, for instance, route planning, and the activation of neuroplasticity.

Most studies focus on the physical effects of therapeutic climbing from a medical perspective on (chronic) back pain, muscle endurance, heart rate, VO_2_max or body flexibility, e.g., [23,24,25]. Ref. [26] conducted an RCT intervention study with 27 persons with multiples sclerosis to evaluate the effects of therapeutic climbing. The results show significant effects for both physical and mental health, namely: on stability, health-related quality of life, and social dimensions, such as social acceptance, one’s ability to contact with others, self-assurance, and mood. Ref. [27] compared therapeutic climbing to other physiotherapy interventions for people with multiples sclerosis and found it to be an equally effective treatment. 

For people with intellectual disability, no research can be found. Ref. [28] described the possible effects of climbing on the target group theoretically and [29] constituted a learning concept for people with intellectual disability in 1999, which has not been evaluated yet. Therefore, there is a conspicuous lack of research in the field of physical, psychological, and social effects of climbing on people with intellectual disability and people with disability in general. In addition to this, the focus has been on therapeutic climbing and its effects; research conducted on the effects of a wider variety of sports climbing has not yet been conducted. This also includes qualitative data in order to explore new phenomena in climbing and its effects. Qualitative research should be conducted in order to assess effects that have not yet been anticipated. This research gap has to be filled.

## 2. Aims, Materials and Methods

The research project ‘Effect of Indoor Climbing on the Occupational Self-Efficacy and Employability: Results of a 10 Months Randomised Controlled Study of Persons with Intellectual Disability’ is the first study identifying the benefits of indoor climbing as a measure to increase the employability of people with intellectual disability in a sheltered workshop. Furthermore, possible positive results of this study could help to support the contribution of the transition of people with intellectual disability into the general labour market. This project took place over the period from 1 January 2018 to 30 September 2019.

Since overcoming limits and achieving success in climbing can mainly be attributed to one’s own abilities [26], the influence of climbing on self-efficacy was also investigated within the research project. In the professional context, self-efficacy has a decisive influence on career decisions and choices, as well as with regard to professionally relevant interests, values, and goals [30]. In order to be able to present the possible effects of indoor climbing better on a comprehensive basis, the effect on self-perception and work performance of employees with intellectual disability in a sheltered workshop were also examined as part of this research project. As a result of this background, four research questions were established, in which only the following two questions arose for this project: What influence does indoor climbing have on the employability of employees with intellectual disabilities in a sheltered workshop?What influence does indoor climbing have on occupational self-efficacy?

### 2.1. Study Criteria and Design

To be able to answer the two research questions mentioned above, Faul’s G*Power test. version 3.1 from Düsseldorf, Germany, was carried out first to compute the total population size [31]. To do so, an effect size of 0.5 was chosen, which represents a medium effect [32]. The results showed that a total of 192 participants were needed for this study to investigate the abovementioned research questions. At the end of the estimation of the total population, an additional 10% drop out rate was added to ensure the minimal number of participants. Thus, the main goal for this project was set on a total of N = 210 participants. The number of participants should be allocated over the sheltered workshops as equally as possible. Therefore, a sheltered workshop in Germany could participate in the study if they were able to find at least 30 participants. This number was set up to assure the minimum of ten participants in each of the constructed groups. Therefore, the group size was also equal among the different sheltered workshops and thus each of the defined groups reached a minimum of 70. Out of the 13 sheltered workshops contacted, ultimately seven participated in the study. Six of these sheltered workshops were located in the state of North Rhine-Westphalia and one in the state of Mecklenburg West Pomerania. The study’s admission criteria for participants were as follows: they needed to be employed at a sheltered workshop, between 18 and 65 years old, and diagnosed with an intellectual disability. In Germany, sheltered workshops are special designed for persons with severe disabilities to take part in daily working life.

The study design was an experimental study with a randomised controlled study (RCT) design, which included one intervention group and two control groups. Furthermore, participants were randomly allocated, using a randomising programme, into one of the three groups. Participants in the intervention group (IG) went indoor climbing two times a week for two and a half hours under the supervision of indoor climbing trainers and individual assistants. During the sessions, participants were free to climb and no main goals were set in advance or at the beginning of each session. The main reason of doing so was to give the participants free choice and not to set objectives and therefore creating internal and external pressure. Furthermore, IG participants received climbing equipment for free, including a climbing harness and shoes and their entrance fees were funded. The first control group (CG1) followed a sports programme two times a week for one and a half hours, which was also funded by the project. The sports programme could be completed in a sports hall at their own sheltered workshop or at a recreation centre nearby. Each CG1 was free of choice to select a joint sport program, such as indoor football, rhythmic dancing or courses at a fitness studio. Lastly, the second control group (CG2) served as controls and were not offered additional sports programmes. The project duration was 10 months in total, the reason of choosing this time period was twofold. First, a rather sustained time period was selected to assure that the intervention can cause possible effects. Second, after the study’s completion, participants from both control groups CG1 and CG2 were also given the opportunity to go indoor climbing for 10 months, for which they received the same equipment and conditions as the original IG. In this way, the participants time of waiting was not disproportionate and the risk of drop out due to motivational problems will not be too high. Next to that several organisational reasons needed to be considered.

Since the project’s main focus was to investigate the effect of climbing on subjective constructs, no objective measurements, such as physical performance tests, were conducted. All participating people with intellectual disability were asked about occupational self-efficacy and employability at the beginning (t1) and after the intervention (t2), with a questionnaire using closed questions.

### 2.2. Instrumentation

Ref. [14] developed a scale with 26 items which measure *employability* on the basis of six subscales, which are the following: (1) job-seeking and continuing education, (2) motivation, internal and external combined, (3) willingness to make concessions, both on labour and merit, (4) cognition of one’s abilities, (5) social conditions, and (6) mental and somatic health. For the current study, 36 additional items [33] were added to one of the subscales of the employability scales used in [14]. A total of 19 additional items were created by the authors themselves and were partially used as control questions.

Ref. [34] developed a scale to measure the *occupational self-efficacy*. The scale measures the persuasion of someone’s behaviour in an occupational context and consists of 19 items. The items are rated on a 6-point Likert scale. The occupational self-efficacy scale consisted of 19 items and the scale showed scores between excellent and good for internal consistency. Cronbach’s alpha was 0.92 and the split-half reliability after Spearman–Brown was 0.88. Furthermore, construct validity is given.

### 2.3. Modifications for People with Intellectual Disability

Questioning people with intellectual disability using standardised questionnaires with closed answers requires certain adjustments. Questions have to be easy to understand, should avoid negations, and should not contain technical terms or quantitative valuations. Therefore, the adjustments mentioned above were applied to all questionnaires and these questions were translated into easy-to-understand language. Furthermore, in order to improve the items’ accessibility, the ones concerning occupation were adjusted to the sheltered workshop environment. Previous studies have shown higher levels of systematic answering tendencies such as acquiescence or recency effect [35,36,37,38].

As it is essential (and compulsory) to question people with intellectual disability, so that they can speak for themselves, it was necessary to resolve certain modifications. In order to reduce complexity and quantitative valuation a 3-point scale is recommended [39] and was used for the current study: ‘Yes’, ‘Somewhat’, and ‘No’. Even though this scale is limited, previous studies have shown it is best used for people with intellectual disability, since more categories require a higher ability of abstraction, which cannot be expected generally and could result in missing data. In addition, at the beginning of the interview, all response options were shown to the participant, printed out on different cards and explained to the participant in detail. These response options, as well as the whole questionnaire, were developed and discussed beforehand with people with intellectual disability. Notably, the questionnaires were also conducted in the form of an interview. By doing this, each participant was able to understand the items of the questionnaire in the best way possible and no reading competence was presupposed. 

Several studies have shown that most of the participants with intellectual disability tend not to answer unknown or difficult items [39]. This causes an enormous information deficit, since data of items are missing and thus no mean can be generated for these participants. Consequently, most studies can make no reliable prediction. To overcome this difficulty, a method was developed by [39] to prevent large amounts of data loss in studies with participants with intellectual disability. The method exchanges the values of missing items by the mean of the remaining answered items. However, this procedure is limited to a maximum of twenty percent of missing items of the total questionnaire and the use of this method leads to less missing values; therefore, the information of more participants can be included.

### 2.4. Procedure

Interested sheltered workshops received flyers with detailed information about the project in accessible language, which were distributed among the employees with intellectual disability. Employees interested in participating informed their supervisor who signed them up.

To certify the participants’ data privacy, the lists with their full names were encoded. The interviews were held in a separate office space at the sheltered workshop and each participant was informed about the purposes of the interview, the interviewing procedure, and the response options. Finally, after all participants were interviewed, the two groups could start with their specific programme, the intervention or sports programme.

### 2.5. Analyses

All reported *p*-values are from two-sided hypothesis tests. Statistical significance was defined at a ≤0.05 level. All statistical analyses were performed by [40]. ANOVA tests with Scheffe Post Hoc tests for multiple comparisons were used to analyse differences between age, gender and sport intervention groups. Furthermore, *t*-tests were carried out to analyse changes over time. Possible effect sizes were determined by using Cohen’s d deviation and interpreted further to determine its relative size [41].

## 3. Results

Figure 1 displays a CONSORT diagram and shows the study enrolment [42]. Out of 212 participants, three participants refused to participate after hearing the first questionnaire, due to either parental prohibition or lack of motivation. After t2, 185 participants were included in the study, thus obtaining a dropout rate of 12.7% (27 participants (13 = due to lack of motivation; 6 = moved to different city; 3 = had an injury; 3 = social problems; 2 = other)).

### 3.1. Descriptive Analysis

There was a gender imbalance among the participants: the number of men (71.4%) in the study being almost three times higher than women (28.6%). The age ranged between 18 and 62 years (*M* = 33, *SD* = 11.20) and most participants were aged between 18 and 29 years, which covered almost half of the study’s population. More than a quarter of the population was between 30 and 39 years old. 

Furthermore, both participants from the age groups 40–49 years and 50 years and older combined represented only a quarter of the sample size (see Table 1).

### 3.2. Correlational Analysis

In the following section, the results of the different indicators are explained separately. For these indicators, the representation of the results can show different sample sizes, as not all of the participants gave answers to all questions consistently. A twenty percent procedure was executed to increase the number of participants for each indicator (see Section 2. Aims and Method).

At t1, data for *employability* showed no significant differences between the different groups, age, and gender. After the intervention, the mean for employability stayed almost the same and therefore no effect over time was visible. A 3 × 2 repeated measures ANOVA revealed no significant group and time effect for employability regarding the three groups. Differences between age and gender did not occur as well. *t*-tests showed significant increase in mental and somatic health over time, from t1 (*M* = 1.45, *SD* = 0.28) to t2 (*M* = 1.53, *SD* = 0.26) for the IG (*t*(53) = −2.42, *p* = 0.019, *d* = 0.33) but no change in both CG 1 and CG 2. In addition, the data showed no differences in age and gender for mental and somatic health. 

The mean of *occupational self-efficacy* at t1 showed no significant differences between the three groups and age. However, a significant difference between gender could be found. Females had a significant lower mean of occupational self-efficacy (*male: M* = 1.42, *SD* = 0.32; *female: M* = 1.30, *SD* = 0.34) than men. At t2, mean decreased for the whole sample size from 1.41 to 1.39, although the change was not significant. A 3 × 2 repeated measures ANOVA revealed no significant group and time effect for self-efficacy regarding the three groups. Neither could any age differences been found, but there were differences in gender. The mean value of self-efficacy of the female participants from the climbing group increased significantly (*t*(15) = 2.07, *p* = 0.039, *d* = 0.52) from t1 (*M* = 1.28, *SD* = 0.36) to t2 (*M* = 1.39, *SD* = 0.31). The mean values of female participants in CG1 and CG2 decreased over time. Overall, the gender gap in occupational self-efficacy (*F*(1,164) = 7.47, *p* = 0.007, *η^2^ p*= 0.044, *n* = 176) is still significant at t2 but decreased over time. The effect size according to [41] was *f* = 0.21 and thus corresponds to a weak effect.

## 4. Discussion

The results of the presented study show the effects of indoor climbing on different levels and in different areas in the life of people with intellectual disability. To increase the overall employability of people with intellectual disability in sheltered workshops, data clarify that indoor climbing does not have a significant effect on all aspects, but it does on mental and somatic health, which is listed as one of the key subscales of employability (Hypothesis 1). Therefore, the outcome only partially supports Hypothesis 1. Several studies have demonstrated that the psychosomatic health of people with intellectual disability is concerning. For example, Ref. [43] performed a review (until 2002) and stated that between ten and eighty percent of people with intellectual disability suffer from mental health problems. Ref. [43] clarifies that this broad range mainly depends on several reasons, such as methods of case identification and the population studied, and definitions of mental health problems. Furthermore, in the literature review (up to 2007) carried out by [44] it is also highlighted that the majority of people with intellectual disability suffer from severe and enduring mental ill-health. In both literature reviews by [43,44], the lack of development of psychosocial interventions for people with intellectual disability becomes abundantly clear. These findings are congruent with [15]’s statements about the psychosomatic health status of persons with intellectual disability. The authors state that psychosomatic health displays a positive influence on employment probabilities of persons with intellectual disability, which, however, in most cases, is lacking. The study presented showed a significant increase for mental and somatic health over time for the IG. Subsequently, the utility of implementing indoor climbing as a regular activity in their weekly work routine can be twofold. First, it shows that climbing can strengthen both mental somatic and physical health, since climbing contains complex movements [22]. Next, indoor climbing can contribute to the development of employability to either expand, maintain, or take up employment [5]. Therefore, indoor climbing could be established in addition to other concepts, in order to develop work-related competences and abilities, represented by other subscales of employability: for instance, negotiating traffic independently, reading competences, further education, and decision making.

The current study showed that the effect of indoor climbing is limited for self-efficacy (Hypothesis 2). The results displayed a gender difference for self-efficacy, in which men exhibit higher levels of self-efficacy for both data points. Previous research does not consistently show gender differences of people without disability for self-efficacy, e.g., [45]. However, in [46] it is emphasised that gender differences can play a role for self-efficacy in the working environment. Furthermore, it was demonstrated that gender differences were only significant in occupational self-efficacy and no significant differences were found for the other dimensions involving emotional and social contexts. In this current study, it was also asserted that women with intellectual disability sense they have a lower occupational self-efficacy than do men with intellectual disability. Ref. [47] emphasised that women lack an overall involvement in decision-making and that this may be a result of gender discrimination in the Western community. Ref. [48] investigated the relationship of occupational self-efficacy with gender and point out that a significant gender difference for occupational self-efficacy is given when the occupation is categorised as a male-dominated occupation. In addition, the authors point out that gender is not a reason that leads to occupation differentiated by gender alone, but is based on the aspects of gender role socialisation in our society. The participants’ occupations in the current study were operated in warehouses with lumber and metal. In addition, the gender distribution in these warehouses was disproportionate, largely in favour of men. These occupations can be combined and seen as engineering occupations, which is clustered as a non-traditional occupation [49]. Furthermore, they revealed that only thirty percent of women felt suitable to become an engineer. As a result, the study demonstrated that woman had low expectations of non-traditional occupations, resulting in occupations which are male-dominated [49]. The result of the current study supports these findings, with women exhibiting lower levels of occupational self-efficacy. However, the results of the current study further revealed that women’s level of occupational self-efficacy rose significantly over time; thus, only women with intellectual disability benefitted from indoor climbing. Therefore, according to the data of the study presented, indoor climbing is a reasonable method for women with intellectual disability to improve their self-efficacy and increase belief in their own competences and abilities in the working environment. Thus, indoor climbing can help to reduce the gender gap in occupational self-efficacy of people with intellectual disability. Since fewer women with intellectual disability wanted to take part in this study than men did, further research is necessary to determine specific reasons for the results, such as different types of occupation, and to be able to confirm these findings. Next to this, further research is needed to investigate the effect of indoor climbing on dimensions of self-efficacy with regard to social and emotional factors.

### Strengths and Limitations

This is the first study involving people with intellectual disability to show the effects of indoor climbing in an RCT design. Conducting a study with an RCT design is rare, especially one which involves more than 200 people with intellectual disability, since including them in research comes with barriers like no or limited reading competences, a requirement for the use of easy language, special living environments (e.g., little or no access to the Internet), a low capacity for remembering, and a low ability to abstract. Therefore, research with people with intellectual disability is more time consuming and requires more effort, which is why sample sizes are relatively small in general and hardly achieve the necessary sizes in order to use statistical procedures, in particular ones relevant in interference statistics. In addition to this, very few validated instruments currently exist to question people with intellectual disability. For this reason, prior to the beginning of this study the instruments used were discussed with randomly selected people with intellectual disability, who did not go on to participate in the study. In order to maximise the study’s feasibility, these people were asked for an appraisal of the instrument and the study’s sequence. By doing so, first peer’s reviews concerning the study design were collected and comments could be adjusted in time. 

Furthermore, in addition to climbing equipment and the entrance fee, transportation to and from the climbing gym was funded for all participants. It became apparent that this was very valuable for the study’s success and was a positive influence on the participant’s state of mind. Transportation was organised at fixed times and this led to an additional stability in the participants’ daily work structure. Participants’ personal responsibility was encouraged in order that they would arrive on time and bring all the necessary personal climbing equipment. In addition, during the journeys participants exchanged views about their climbing performance and got to know each other further. Over time, several friendships developed among the participants particularly as a result of the group transportation.

Therefore, not only are the results of this study relevant for supporting people with intellectual disability in work related aspects, but the procedures and instruments developed within this study are valuable for further research with people with intellectual disability.

This study also has limitations. Participants were mostly selected from one state of Germany and the distribution of gender in the actual sample was not in accordance with that of sheltered workshops, 41 per cent female, in Germany [7]. Furthermore, all participants were diagnosed with an intellectual disability and therefore they were allocated to a sheltered workshop nearby. However, further participant characteristics such as the level of disability was not administered during the project. As a consequence, additional research outcomes of the influence of climbing on the different levels of intellectual disability is absent. Additionally, the influence of climbing itself on the participants’ emotional state, such as fear of falling, stress, anxiety, happiness or relaxed, has not been investigated before, during and after the intervention. 

In general, questioning people with intellectual disability with standardised questionnaires is still affected by certain limitations. Even in easy language, after pre-tests and discussions with people with intellectual disability we cannot be sure about every participant completely understanding all of the questions. In addition, the three-point scale is limiting in analyses. Nevertheless, it is still the best way to achieve as little missing data as possible. With regard to these limitations further methodological research is needed and more experience with standardised questionnaires applied to people with intellectual disability is both valuable and necessary.

## 5. Conclusions

Up until this study, almost all participants were unfamiliar with climbing as a sport. At the end of the study, all participants showed positive attitudes towards climbing and were stimulated by the overall attractiveness of it. In addition, the participants also proudly informed their peers at the sheltered workshops about their experiences and therefore, to an extent, were infectious in their enthusiasm about climbing during the study. One possible reason of the increased popularity of climbing nowadays is that there are several types of climbing, which are based on an innovative perspective [50]. Not only is climbing an increasingly popular sport, but it can also be a valuable catalyst for improving the psychosomatic health of people with intellectual disability as well as improving women’s occupational self-efficacy. This study adds further findings to previous studies, which have shown that climbing has positive effects on back pain, straighter posture, and can help individuals to feel stronger and fitter overall.

The findings of the study presented indicate that people with intellectual disability can widely benefit from climbing. Yet, research in the field of recreational climbing with people with (intellectual) disability is still at the very beginning. Therefore, further research is necessary to uncover specific benefits from climbing for people with intellectual disability. It would be of high interest to investigate the effects of climbing of certain groups, such as woman only, since these were underrepresented in the study, but yet benefitted more than men. In addition, this could also be the case for special age groups. In particular, age groups who are right before coming-of-age would be of interest, since they can adapt the effects of climbing on an early stage of life and benefit from the adapted social skills. Next to this, further research of possible long-term effects of climbing in comparison to other physical activities can provide new insights on several levels, such as emotional and social, and also on the participation in different areas of life of people with intellectual disability. Moreover, climbing is a versatile sport, since it includes many kinds of climbing based on innovative and modern perspectives. Therefore, climbing can demonstrate further novel perspectives for future studies and promote the effects of climbing on people with and without (intellectual) disability to a wider extent.

Overall, the results of the current study have shown that indoor climbing can be an effective method for people with intellectual disability who work in sheltered workshops, not only to increase peoples’ health, but also to increase work relevant competences and attitudes, too.

## Figures and Tables

**Figure 1 ijerph-19-13399-f001:**
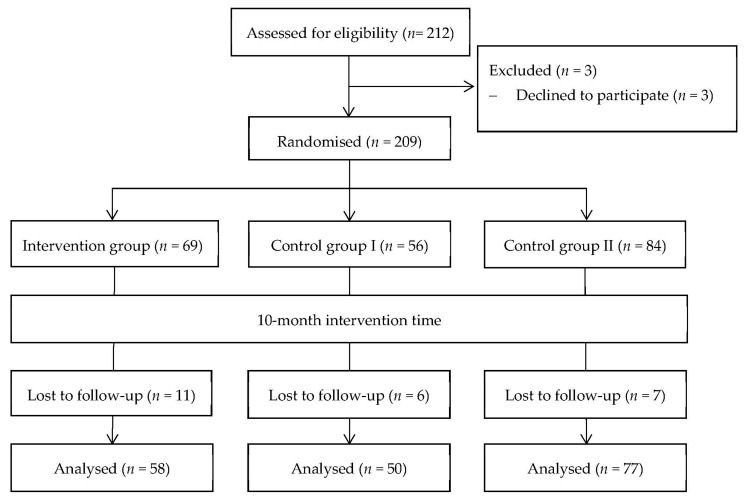
CONSORT flow diagram of participants included and excluded during the course of the project.

**Table 1 ijerph-19-13399-t001:** Descriptive data of the sample (*N* = 185).

Baseline Characteristics	*n*	%
Group		
intervention group	58	31.4
control group I	50	27.0
control group II	77	41.6
Gender		
female	53	28.6
male	132	71.4
diverse	0	0
Age group		
18–29 years old	89	48.1
30–39 years old	50	27.0
40–49 years old	23	12.4
50 years and older	23	12.4

## Data Availability

Some or all data and models that support the findings of this study are available from the corresponding author upon reasonable request.

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
