# Peer review of "Effect of Indoor Climbing on Occupational Self-Efficacy and Employability: Results of a 10-Month Randomised Controlled Study of Persons with Intellectual Disability"

_ijerph, 2022, doi:10.3390/ijerph192013399_

Round 1

Reviewer 1 Report

General comments

Effect of Indoor Climbing on the Occupational Self-Efficacy and Employability: Results of a 10 Months Randomised Controlled Study of Persons with Intellectual Disability

This article the definition of the research topic, comprehensive literature review, selection of studies, data collection, analysis of data, and reporting of results were done appropriately. In addition, the authors have done a great job of providing an informative and meaningful addition to the current study field. However, there are several changes that the authors are encouraged to revise to elevate the overall contribution of the paper to this research field.

Line 12: randomised controlled study (RCT), write a description of the abbreviation

Line 144: (Study criteria and design)

-          How the intellectual disability levels of the subjects were evaluated?

-          intervention group (IG) indoor climbing about more explain (about the training program

-          The first control group (CG1) more explanation about training programs (running, walking, etc.)

-          Was the physical performance of all participants measured during the study?

-          Were the acute emotional states of the intervention group examined during climbing? (fear of falling, anxiety, etc.) if not evaluated add it to the limitation section.

-          Line 227: provide more information about the analysis methods used in the study (Correlational Analysis, T-tests, Anova, effect size, etc.)

Line 230: check the sentence

Line 245: check the sentence

Author Response

Dear reviewer,

thank you for your honest and helpful review. Please see the attachment, in which I explained your points separately. I found all of the points made very useful and I do hope that I could rewrite these in the best possible way.

Thanks again for your time.

Kind regards,

The authors

Reviewer 2 Report

Dear Authors,

It is a pleasure to review this paper, it looks promising and interesting from many angles of view, however, it should be significantly improved. My suggestions:

1. Please revise your abstract, so it can be more engaging and coherent (even with the word count limit). Probably, the opening sentence (9-10) will sound more logical in this form: "Indoor climbing can have has different effects on various levels, including a physical, psychological, and social level. "

2. the beginning of the introduction may raise a discussion, hence, needs to be explained. As far as I understand, climbing as a sport takes its beginning in the 18-19 centuries. If you mean certain "new" kinds and sub kinds of climbing, it needs to be explained well, without general delusive sentences (23-25). 

3. (44) Bundesarbeitsgemeinschaft Werkstätten für behinderte Menschen should be translated and explained in English, maybe in footnotes...

4. How the size of the same (N=30) has been defined? Is it sufficient to achieve this study's goals? 

5. Why the project lasted 10 months? How would you explain this duration? Please include it in the paper.

6. There are several errors detected in results section, please fix it.

7. (58) a space is missing

8. (372-381) limitations can be extended and explained in detail.

9. (395-398) The future research directions are poorly proposed and are not outlined and explained. Authors present this topic as underresearched and perspective, and a better focus should be done on future research directions and perspective angles of the current study. It would be a logical Extention of the limitations part. 

Author Response

(The authors gave the same response as above.)

Round 2

Reviewer 2 Report

Dear Author(s),

Thank you for your efforts. All my recommendations have been addressed, and the paper looks significantly improved. However, there is still room for further revisions. A few more suggestions to polish your piece of work:

1. (268-269) the sentence seems messy, could you please revise it? 

2. (450) tautology: further... please rephrase

3. As for the future and prospective research directions, I would propose to attract a reader's attention to new certain kinds of climbing as a new innovative sport from the innovation perspective, in the spirit of Ekaterina Glebova & Michel Desbordes, 2021. "Technology innovations in sports: Typology, nature, courses and impact," Chapters, in: Vanessa Ratten (ed.), Innovation and Entrepreneurship in Sport Management, chapter 5, pages 57-72, Edward Elgar Publishing. It can be mentioned at the beginning of inclusion, and later, linked to further research suggestions, then demonstrate new angles of view for future studies. 

Author Response

Dear reviewer,

thanks again for your useful review. Your input was very helpful, especially the proprosal. I do hope that I could revise your points in such a manner and that they are up to standard. 

Thanks again for your time.

Kind regards,

The authors

Round 3

Reviewer 2 Report

Dear Authors,

Thank you for addressing all recommendations.